# Nocturnal Pain Is Not an Alarm Symptom for Upper Gastrointestinal Inflammation but May Be an Indicator of Sleep Disturbance or Psychological Dysfunction

**Jacob Cindrich [1], Chance Friesen [2], Jennifer Schurman [3], Jennifer Colombo [3]** and **Craig A. Friesen [3],***

[1] University of Kansas School of Medicine, Kansas City, KS 66160, USA; jcindrich2@kumc.edu
[2] Division of Gastroenterology, Hepatology, and Nutrition, Children's Mercy Kansas City, University of Kansas School of Medicine, Kansas City, KS 66160, USA; csfriesen@cmh.edu
[3] Division of Gastroenterology, Hepatology, and Nutrition, Children's Mercy Kansas City, University of Missouri-Kansas City School of Medicine, Kansas City, MI 64108, USA; jschurman@cmh.edu (J.S.); jmcolombo@cmh.edu (J.C.)
* Correspondence: cfriesen@cmh.edu

**Abstract:** Alarm symptoms are widely used in pediatric gastroenterology to discern when abdominal pain needs further workup. Despite wide use, the data supporting the validity of these symptoms are not well established. This study explored one alarm symptom—nighttime waking with pain—and its associations with histologic inflammation of the upper gastrointestinal tract, psychological dysfunction, and disordered sleep. This retrospective study evaluated 240 patients with abdominal pain-related disorders of the gut–brain axis (AP-DGBI). Patients underwent questionnaires related to sleep disturbance, behavioral assessment, and gastrointestinal symptoms, including Rome IV criteria for AP-DGBI. Routine upper endoscopy with standardized biopsies was performed in 205 patients. Endoscopy results showed no association between esophageal, gastric, or duodenal histologic inflammation and nighttime waking with pain. Nocturnal pain was associated with increased scores for both psychological and sleep disorders, including social stress, depression, disorders of initiation and maintenance of sleep (DIMS), disorders of daytime somnolence (DOES), and sleep hyperhidrosis (SHY). This study concluded that nocturnal pain is not a reliable predictor of upper gastrointestinal inflammation but may be a prognosticator for psychological distress and sleep disturbances.

**Keywords:** abdominal pain; functional dyspepsia; irritable bowel syndrome; anxiety; sleep disorders; inflammation

## 1. Introduction

Chronic abdominal pain is common in children and adolescents, with a prevalence of 19 to 38% [1,2]. Most of these youth fulfill criteria for an abdominal pain-related disorder of gut–brain interaction (AP-DGBI) as defined by Rome IV criteria [3]. The two most common AP-DGBIs are functional dyspepsia (FD; defined by the presence of epigastric pain or burning, early satiety, or postprandial fullness) and irritable bowel syndrome (IBS; defined by a change in stool frequency, a change in stool form, or a change in pain level with defecation) [3]. These diagnoses are symptom-based and, although they require that symptoms cannot be explained by another medical condition after appropriate evaluation, they are intended to be positive diagnoses not requiring tests to rule out recognized organic diseases [3].

There are no established, evidence-based guidelines for the evaluation of youth with chronic abdominal pain which creates variability in the work-up of these patients with widely varying costs across health systems [4,5]. Endoscopy is a common diagnostic procedure in children and adolescents with dyspepsia and has been shown to effect patient management with a positive effect on outcomes [6,7]. In a systematic review, Thakkar and colleagues reported on 18 studies, 15 of which included histologic diagnoses [8]. The rate

of histologic inflammation varied from 23% to 93%, indicating different disease prevalence at different healthcare sites and/or differing selection criteria for endoscopy [8]. Although the clinical significance of some forms of inflammation is unknown and other forms do not have well agreed upon criteria, the current practice is to utilize endoscopy to assess for the presence of organic disease [9–13].

For pediatric patients fulfilling criteria for functional dyspepsia in particular, it is common to initially treat patients with acid reduction therapy and then proceed with endoscopy in non-responders [14]. As this test is invasive and costly, there has been an attempt to identify predictors of diagnostic yield to be more selective of patients who undergo endoscopy. A common approach is to use alarm or "red-flag" symptoms as an indicator, though the evidence for this approach is not robust. These alarm symptoms are symptoms that may increase the pre-procedure probability of finding an organic disease [15]. Alarm symptoms are used in medicine to help differentiate true, urgent pathology from clinical diagnoses that can be made largely without testing. In the world of pediatric abdominal pain, these same approaches have been trialed to demarcate organic diseases such as inflammatory bowel disease (IBD), peptic ulcer disease, and esophageal disease from their counterparts, the AP-DGBIs. In practice, a lack of alarm symptoms is used as an indicator for omitting invasive tests to rule out organic disease. In actuality, the utility of these alarm symptoms is not clear. Multiple studies have shown no significant difference between the presence of alarm symptoms in patients with organic versus functional disease, while others have [7,16–20]. With the exception of IBD, there is minimal pediatric literature supporting the alarm symptoms themselves [21,22].

Nighttime waking with pain (nocturnal pain) is one commonly utilized alarm symptom used as an indicator for upper gastrointestinal tract endoscopy [23–27]. Previous studies have reported mixed results on the utility of nocturnal pain as an alarm symptom [16–19,28]. However, only two studies have assessed and reported histologic findings. One study utilized an array of alarm symptoms as an indication to perform endoscopy [7]. Pre-selecting patients by way of alarm symptoms may limit what correlations can be made. The other study found the diagnostic yield to be 32% in those with nocturnal pain as compared to 7.5% in those without nocturnal pain; however, only esophagitis and H. pylori gastritis were considered organic diseases [28].

The primary aim of the current study was to evaluate the singular symptom of nocturnal pain as a predictor of histologic inflammation in a group of youth with FD (with or without overlapping IBS) who underwent upper endoscopy due to a lack of response to acid-reduction therapy, a common pathway to endoscopy that is not driven by the presence of alarm symptoms. A secondary aim was to assess the relationship between the presence of nocturnal pain and psychologic and sleep dysfunction.

## 2. Results

A total of 240 patients were included in the study. They ranged in age from 8 to 18 years with a mean age of 13.2 ± 2.6 years and 73% were female. The percentage of patients fulfilling the specific Rome IV criteria is shown in Table 1.

**Table 1.** Rome IV diagnosis as a percentage of the overall cohort and prevalence of nocturnal pain for each Rome IV diagnoses.

| Diagnosis | % of Total Cohort | % Reporting Nocturnal Pain |
|---|---|---|
| FD only | 39 | 65 |
| IBS only | 9 | 40 |
| Both FD and IBS | 52 | 53 |

FD = functional dyspepsia; IBS = irritable bowel syndrome.

Overall, nocturnal pain was reported by 56% of patients. There was no difference in the prevalence of reporting nocturnal pain comparing females to males or children (<13 years of age) to adolescents (≥13 years of age), respectively. The percentage of patients reporting

nocturnal pain for each Rome IV diagnosis is shown in Table 2. There were no significant differences in the prevalence of nocturnal pain between diagnoses. Nocturnal pain was most associated with upper abdominal pain. The percentage of patients reporting pain at each specific site comparing those with and without nocturnal pain are shown in Table 2.

**Table 2.** Comparison of percentages of patients reporting pain at different sites comparing those with and without nocturnal pain.

| Site of Pain | Nocturnal Pain | No Nocturnal Pain | *p* Value |
|---|---|---|---|
| Chest (heartburn) | 60 | 54 | 0.368 |
| Upper abdomen | 72 | 48 | <0.001 |
| Epigastrium specifically | 55 | 57 | 0.805 |
| Periumbilical | 57 | 56 | 0.520 |
| Lower abdomen | 58 | 54 | 0.547 |

Overall, esophageal inflammation was seen in 23% of patients (19% GERD and 4% EoE), gastric inflammation in 31% (1% acute gastritis, 21% chronic gastritis, 8% eosinophilic gastritis, and 1% H. pylori-positive gastritis), and duodenal inflammation in 12% (chronic duodenitis in 3%, acute duodenitis in 3%, and eosinophilic duodenitis in 6%). There was no difference in the prevalence of inflammation at any site comparing females to males or children to adolescents, respectively. Nocturnal pain was not reported more frequently by patients with inflammation at any of these sites. (See Table 3). Within the antrum, if chronic gastritis was excluded, nocturnal pain was reported by 24% of those with inflammation vs. 9% of those without inflammation ($p = 0.026$).

**Table 3.** Prevalence of histologic inflammation comparing patients with and without nocturnal pain.

| | Nocturnal Pain | No Nocturnal Pain | *p* Value |
|---|---|---|---|
| Esophagitis | 26% | 20% | 0.575 |
| Gastritis | 29% | 34% | 0.244 |
| Duodenitis | 12% | 11% | 0.105 |

Nocturnal pain was associated with increased scores for social stress, depression, DIMS, DOES, and SHY, but not anxiety (see Table 4). Nocturnal pain was reported by 59% of patients in cluster 1, 42% in cluster 2, and 72% in cluster 3 ($p = 0.003$).

**Table 4.** Scores for psychological functioning and sleep disturbances (mean ± STD) comparing patients with and without nocturnal pain.

| | Nocturnal Pain | No Nocturnal Pain | *p* Value |
|---|---|---|---|
| Social stress | 49.1 ± 10.3 | 46.2 ± 9.5 | 0.026 |
| Anxiety | 54.9 ± 11.9 | 52.9 ± 10.1 | 0.172 |
| Depression | 51.0 ± 9.9 | 47.0 ± 8.5 | 0.001 |
| DIMS | 16.5 ± 5.1 | 15.0 ± 5.5 | 0.048 |
| DOES | 9.9 ± 3.8 | 8.8 ± 3.6 | 0.046 |
| SHY | 3.0 ± 2.2 | 2.5 ± 1.2 | 0.020 |

DIMS = disorders of initiation and maintenance of sleep; DOES = disorders of excessive daytime somnolence; SHY = sleep hyperhidrosis.

## 3. Discussion

In the current study, we found nocturnal pain to be more highly associated with upper abdominal pain and a trend towards an association with FD. Despite this, nocturnal pain demonstrated no association with inflammation, in general, at any of three distinct locations

including distal esophagus, gastric antrum, and duodenum. We did find an association with antral inflammation when chronic gastritis, a condition of unknown significance, was excluded from the analysis. However, this only left 10% of the total sample. We were only able to identify three previous studies reporting on the relationships between histologic inflammation and alarm symptoms [7,16,28]. One of these did not analyze histologic inflammation separate from other organic diagnoses [16]. One only considered esophagitis and H. pylori gastritis as organic disease [28] In the latter, Thakkar and associates found no association between nocturnal pain and histologic inflammation [7]. In their study, all patients had at least one alarm symptom, the criteria used for proceeding with endoscopy. In the current study, we did not require an alarm symptom to be present, just non-response to acid-reduction therapy, and our findings were consistent with those of Thakkar and associates. We concluded that nocturnal pain is not an alarm symptom predicting histologic inflammation, except, perhaps, in the antrum, which needs to be evaluated in a larger sample given the low base rate.

AP-DGBIs are best viewed through a biopsychosocial model which states that symptoms arise through an interaction of biological, psychological, and social factors, which are interactive with each other. Therefore, we further assessed relationships between nocturnal pain and specific sleep disturbances and psychological dysfunction. Nocturnal pain was associated with several sleep disturbances including disorders and initiation and maintenance of sleep (DIMS), sleep–wake transition disorders (SWTD), disorders of excessive daytime somnolence (DOES), and sleep hyperhidrosis (SHY). Patients reporting nocturnal pain had elevated scores for sleep disturbances that coincide with DIMS, DOES, and SHY. Whether these associations represent cause-and-effect relationships or are epiphenomena cannot be discerned within the cross-sectional design of the study. It is certainly feasible that patients wake at night from a sleep disturbance and then begin to experience pain; it is equally feasible that they wake because of pain and therefore are unable to maintain sleep. Elevated scores for sleep hyperhidrosis may have physiological implications as hyperhidrosis is associated with sympathetic overactivity [29]. The common link could be sympathetic hyperresponsiveness, a counterforce that is supposed to be suppressed during sleep. This would be congruent with proposed mechanisms for DGBIs such as IBS, where sympathetic activity has been directly correlated with disease presence and severity [30–32]. Regardless of the pathophysiology, we found that patients with nocturnal pain are more likely to have higher sleep disturbance scores overall.

Lastly, we evaluated the relationship between social stress, anxiety, and depression with nocturnal pain. Patients with nocturnal pain demonstrated higher scores for social stress and depression, but not anxiety. Additionally, when assessing psychosocial clusters, those with more global distress were more likely to report nocturnal pain; in fact, nocturnal pain was reported by three-quarters of these patients. While the relationship between DGBIs and anxiety and depression has been well documented, to our knowledge, the association of nocturnal pain with psychological dysfunction, either specific to anxiety and depression or more broadly, has not been assessed [33,34]. Again, the study design does not allow us to conclude a cause-and-effect relationship. Sleep disturbances have been associated with psychological dysfunction, possibly linked by autonomic disruption or alteration of circadian rhythms [31,35]. However, the relationship between sleep and psychological dysfunction is likely circular, with psychological dysfunction both contributing to, and the result of, sleep disturbances [36]. Ultimately, more work is needed to determine the pathophysiological links between sleep disturbances, sleep dysfunction, and nocturnal pain. This will require prospective studies to determine whether sleep intervention or psychological therapy reduce nocturnal pain. Presently, we can conclude that nocturnal pain is not an alarm symptom for histologic inflammation but instead may be viewed as an "alarm" symptom indicative of disordered sleep or psychological dysfunction. Given the known association between sleep, psychological dysfunction, and DGBIS, nocturnal pain then becomes an important point for evaluation as it identifies other potential treatment targets within a comprehensive treatment plan for the DGBI.

The strength of the current study is the large sample size with systematically obtained data. Additionally, our patient selection did not rely on the presence of alarm symptoms, thus reducing some bias. Lastly, we evaluated associations beyond biological factors. Our study is not without limitations. The cross-sectional study design only allows for the assessment of associations and not cause-and-effect relationships. Our data set also did not allow assessment of the potential effects of race or body mass index. Also, while we utilized a validated sleep measure, it does not necessarily project what would have been discovered with a sleep study or actigraphy. It should be noted that the current study did not address the utilities of alarm symptoms in general but only sought to assess the usefulness of a single potential alarm symptom.

## 4. Materials and Methods

This was a retrospective study evaluating 240 consecutive patients fulfilling symptom criteria for an AP-DGBI. All patients were seen within an abdominal pain clinic and all AP-DGBI diagnoses were assigned by a single board-certified pediatric gastroenterologist. All patients reported pain at least weekly for a minimum of 8 weeks. All patients completed a standard broad gastrointestinal symptom questionnaire including specific symptoms defined within the Rome IV criteria for AP-DGBI. All patients were specifically asked whether they awaken during the night with abdominal pain. The questionnaire was completed within a Red Cap database which was used for analysis in the current study.

As part of routine care, parents completed the Sleep Disturbance Scale for Children (SDSC), which assesses various specific sleep problems [37]. The SDSC provides subscales, three of which were analyzed in the current study: (1) disorders of initiating and maintaining sleep (DIMS), (2) disorders of excessive somnolence (DOES), and (3) sleep hyperhidrosis (SHY). The SDSC has been validated with actigraphy and has well established psychometrics [38].

As part of routine care, patients completed the Behavior Assessment System for Children- Second Edition (BASC-2) [39]. The BASC is utilized widely in both clinical practice and research, demonstrating both criterion-related and construct validity [39]. Standardized T scores for self-report social stress, anxiety, and depression were used for analysis in the current study. Additionally, patients were assigned to one of three clusters utilizing all BASC subscales as previously described [40]. In general, cluster 1 represents subjects with positive emotional and behavioral functioning, cluster 2 represents subjects with isolated anxiety, and cluster 3 represents patients with broad psychosocial dysfunction (i.e., including both internalizing and externalizing issues).

As a part of routine care, 205 patients had undergone upper endoscopy for FD symptoms and non-response to acid reduction therapy. All patients had normal gross endoscopies (excluding ulcers, erosions, and nodularity) and all had a minimum of two biopsies obtained from the distal esophagus, two obtained from the gastric antrum, and four obtained from the duodenum. All biopsies had been assessed by a board-certified pediatric pathologist. The final reported pathologist diagnoses were utilized for evaluation in the current study.

### Statistical Analysis

SPSS version 23 (SPSS Inc., Chicago, IL, USA) was utilized for all statistical analyses. Descriptive statistics are provided. The percentage of patients reporting nocturnal pain was compared between DGBI diagnoses using a Chi-square analysis. The percentage of patients reporting pain at each abdominal location was compared between patients with and without nocturnal pain using a Chi-square analysis. Scores for sleep and psychological subscales, respectively, were compared between patients with and without nocturnal pain by the Student's *t* test. The percentage of patients reporting nocturnal pain was compared between BASC clusters using a Chi-square analysis. The percentage of patients demonstrating histologic inflammation at each site was compared between patients with and without nocturnal pain using a Chi-square analysis. For comparisons between patients with and without nocturnal pain, a power analysis determined that 82 patients per group would provide 80% power to detect a prevalence difference for inflammation of 20% vs. 40% at an $\alpha$ level of 0.05.

## 5. Conclusions

Nocturnal pain does not provide a useful alarm symptom with which to predict upper gastrointestinal tract inflammation. However, nocturnal pain may be an indicator of social stress, depression, and sleep disturbances, all factors which may affect treatment outcomes. Future studies should address the extent to which sleep or psychological interventions decrease nocturnal pain and the extent to which treatment of pain in general reduces nocturnal pain and subsequently improves sleep and psychological dysfunction.

**Author Contributions:** Conceptualization, J.C. (Jacob Cindrich), C.F. and C.A.F.; methodology, J.S., J.C. (Jennifer Colombo) and C.F.; data curation, all authors, writing—original draft preparation, J.C. (Jacob Cindrich), C.F. and C.A.F.; writing—review and editing—all authors. All authors have read and agreed to the published version of the manuscript.

**Funding:** This research received no external funding.

**Institutional Review Board Statement:** Approval of this study was provided by the Institutional Review Board of Children's Mercy Kansas City.

**Informed Consent Statement:** Informed consent for this retrospective study was waived by the Institutional Review Board of Children's Mercy Kansas City.

**Data Availability Statement:** Data is available upon request.

**Conflicts of Interest:** The authors declare no conflict of interest.

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
