# Peer review of "Nocturnal Pain Is Not an Alarm Symptom for Upper Gastrointestinal Inflammation but May Be an Indicator of Sleep Disturbance or Psychological Dysfunction"

_gastrointestdisord, doi:10.3390/gidisord5030025_

Round 1

Reviewer 1 Report

I would like to congratulate you on this study that I found very interesting. Generally I do not have objections, please check references and language style (I found a sentence in line 213 to 214 that I would like you to consider to arrange  it differently).

Very good English, maybe some minor style changes.

Author Response

With regards to the comments from Reviewer:

  1. We have rewritten lines 213-214, now 207-208, hopefully with increased clarity.

Reviewer 2 Report

The authors present a case series of nocturnal pain in pediatric patients with upper gastrointestinal inflammation. However there are some important issues that need to be adressed:

- first of all a power analysis is missing and this should be calculated and provided within the methods and results section in order to assess if the study is powered for this analysis

- the issue of hand is better addressed by a propensity matched analysis and hence we recommend that one is used based on baseline characteristics that can introduce bias in this study. Better than a cohort study.

- the methods should follow the introduction and not at the end of the manuscript and the introduction itself is too long and it should be shortened,  

Only minor spelling mistakes should be addressed 

Author Response

With regards to the comments from Reviewer:

  1. We have added the power analysis to lines 253-255.
  2. We are certainly willing to do further analysis but would ask for some further direction. I am familiar with utilizing propensity matched analysis for treatment outcome data but I am not clear as to how to set up that analysis in this situation. Also, we enrolled consecutive patients presenting to the clinic and did not otherwise have a selection bias. We did go back and analyze nocturnal pain prevalence pathology findings comparing patients by gender and age group and have included those findings on lines 98-100 and lines 114-115. Again, we would be happy to reanalyze the data with some specific recommendations.
  3. The methods were after the Discussion in the template provided by the journal. We significantly shortened the introduction, creating summary sentences for some portions and eliminating other portions to improve the focus of this section.

Reviewer 3 Report

Thank you for this opportunity to review this manuscript. This study was a retrospective study including 240 patients with abdominal pain-related disorders of the gut-brain axis (AP-DGBI) as defined by Rome IV criteria. The overall purpose of this study was to examine the relationships of Nighttime waking with pain (nocturnal pain) with histologic inflammation, self-reported psychologic and sleep dysfunction. The main findings found that nocturnal pain was Not associated with upper gastrointestinal inflammation, while associated with psychological distress and sleep disturbances. This topic is of interest to AP-DGBI community and is appropriate to this journal. Overall, this is a nicely written manuscript. I have the following suggestions to improve the clarify of quality of the manuscript.

Introduction

1.       The introduction could be more concise, particularly in explaining the significance of the study's purpose. For example, the authors did not mention “nighttime waking with pain (nocturnal pain)” until the final paragraph of the introduction.

2.       It would be beneficial to the readers if the authors would provide a short summary regarding the overall purpose of the stud and its specific aims at the end of the introduction.

Results

1.       Instead of presenting descriptive text results, it would be beneficial to include a table showing participant demographic and clinical variables. This will make it easier for readers to review and interpret the data.

2.       The authors should consider there are additional demographic and clinical factors impacted the study findings, such as race, age, gender, BMI, and medications. These factors should be taken into account when examining the relationships of interest.

Materials and Methods

1.       It would be helpful to include a study flowchart to demonstrate the sample's representativeness within the target population.

Author Response

With regards to the comments from Reviewer:

  1. We significantly shortened the introduction, creating summary sentences for some portions and eliminating other portions to improve the focus of this section.
  2. We added a paragraph laying out specific aims on lines 82-87.
  3. We created 2 new tables (Table 1 and Table 2), moving clinical data from the text.
  4. We added an analysis of nocturnal pain prevalence and pathology findings comparing patients by gender and age group and have included those findings on lines 98-100 and lines 114-115. Unfortunately, our data source does not contain race of BMP data- we added this to the weakness section on lines 197-198.
  5. We just enrolled consecutive patients presenting to clinic so do not have a flowchart to present- any selection would have occurred before referral.

Round 2

Reviewer 2 Report

The authors have addressed the major comments from the previous version

Reviewer 3 Report

The authors have made significant improvements to the manuscript by appropriately addressing the prior comments from the reviewers.